# Seasonal Accumulated Workloads in Collegiate Women’s Soccer: A Comparison of Starters and Reserves

**DOI:** 10.3390/jfmk7010011

**Published:** 2022-01-16

**Authors:** Andrew R. Jagim, Andrew T. Askow, Victoria Carvalho, Jason Murphy, Joel A. Luedke, Jacob L. Erickson

**Affiliations:** 1Sports Medicine, Mayo Clinic Health System, Onalaska, WI 54650, USA; victoria.rvalho@gmail.com (V.C.); luedke.joel@mayo.edu (J.A.L.); erickson.jacob@mayo.edu (J.L.E.); 2Department of Exercise and Sport Science, University of Wisconsin-La Crosse, La Crosse, WI 54601, USA; jmurphy@uwlax.edu; 3Department of Kinesiology and Community Health, University of Illinois at Urbana-Champaign, Urbana, IL 61801, USA; askow2@illinois.edu

**Keywords:** athlete monitoring, workload, GPS, soccer

## Abstract

Research quantifying the unique workload demands of starters and reserves in training and match settings throughout a season in collegiate soccer is limited. **Purpose:** The purpose of the current study is to compare accumulated workloads between starters and reserves in collegiate soccer. **Methods:** Twenty-two NCAA Division III female soccer athletes (height: 1.67 ± 0.05 m; body mass: 65.42 ± 6.33 kg; fat-free mass: 48.99 ± 3.81 kg; body fat %: 25.22 ± 4.78%) were equipped with wearable global positioning systems with on-board inertial sensors, which assessed a proprietary training load metric and distance covered for each practice and 22 matches throughout an entire season. Nine players were classified as starters (S), defined as those playing >50% of playing time throughout the entire season. The remaining 17 were reserves (R). Goalkeepers were excluded. A one-way ANOVA was used to determine the extent of differences in accumulated training load throughout the season by player status. **Results:** Accumulated training load and total distance covered for starters were greater than reserves ((S: 9431 ± 1471 vs. R: 6310 ± 2263 AU; *p* < 0.001) and (S: 401.7 ± 31.9 vs. R: 272.9 ± 51.4 km; *p* < 0.001), respectively) throughout the season. **Conclusions:** Starters covered a much greater distance throughout the season, resulting in almost double the training load compared to reserves. It is unknown if the high workloads experienced by starters or the low workloads of the reserves is more problematic. Managing player workloads in soccer may require attention to address potential imbalances that emerge between starters and reserves throughout a season.

## 1. Introduction

The seasonal monitoring of athletes is becoming a popular strategy across athletic departments and within specific teams. The underlying theory of implementing team-based monitoring is to appropriately manage training-related stress and recovery time throughout a season to optimize performance while reducing the risk of injury and mitigating fatigue as much as possible. The specifics of these relationships are likely heavily influenced by the nuances of the methodology used to quantify workload and what performance-based metrics or indices of recovery are being considered. A consensus statement on athlete monitoring was previously published, which provided guidelines on how to assess workload, and associated parameters to consider when drawing comparisons across different sports, monitoring systems, and software programs [1]. Beyond these specifics, the practicality of each sport setting, the specific physiological demands of the sport and competition schedule likely dictate how a monitoring program could be integrated into regular team activities. A recent systematic review found that athletes were at an increased risk for injury during periods of intensified training, changes in acute training load and following a period of higher accumulated workload [2]. The challenge in applied sport science settings is how to identify the specific thresholds that may constitute as excessive workloads, as they are likely specific to the sport, level of competition (or training experience) and the individual athlete.

Depending on the technology available and specific demands of the sport, various metrics may be evaluated to quantify workloads of athletes. Internal workload is often quantified using objective measures of physiological responses (i.e., heart rate, hormonal fluctuations, training impulses, etc.) [1] or subjective measures such as session ratings of perceived exertion (sRPE), perceived recovery status, perceived soreness, etc. [3]. External workload can be quantified using various measures of movement kinematics, derived from accelerometry, global positioning systems (GPS) [4], or local positioning systems (LPS) [1,5]. Each system produces various metrics and parameters used to characterize movement demands such as total distance, high speed running, movement velocity, acceleration, inertial movement units, and number of sprints. Previous work has reported strong relationships between internal and external measures of workload in various sports [6,7,8,9]. Certain systems even calculate proprietary metrics to further characterize internal and external workload demands incurred by athletes, some of which are summated throughout a session and reflect both volume and intensity of work (i.e., training load by Polar and player load by Catapult). 

Individual sports have unique match demands that are dictated by the rules of play, tactical strategies and fitness level of the athletes. Previous studies have examined the specific match demands across different levels of play in soccer and across each sex [10,11,12,13]. In collegiate women’s soccer, the typical match results in an average distance of ~9800 m, with approximately 1019 m (~10% of total distance) classified as high speed distance. The overall mean velocity of match play was 63 m·m^−^^1^ with an average of 15 sprints per match. This distance and speed of play elicited a heart rate response of 142 bpm or 74% of HR max with peak HR values of 197 bpm equating to ~100% HR max and the mean HR was 74.2 ± 6% HR max [13]. As starters typically play >50% of match time throughout the season, it would be safe to assume they are more representative of the mean HR values and other calculations of match demands that are a reflection of the team’s performance throughout a match. As a result, starters and non-starters are likely to accrue varying workloads over the course of the season. Therefore, it is necessary to analyze HR and workload separately for starters and reserves. 

Currently, limited information is available regarding the accumulation of workloads throughout a season in collegiate soccer and how they differ between starters and reserves. Recently, Curtis et al. [14] were one of the first groups to report on the differences in accumulated workloads between starters and non-starters throughout an NCAA Division I men’s soccer season. While starters accumulated substantially more distance and number of accelerations throughout the season, non-starters accumulated more distance and higher TRIMP volumes during training, indicating that non-starters may complete extra work or conditioning during practice compared to starters. Such discrepancies throughout a season may pose challenges to coaches regarding the management of workloads in starters while also providing an adequate and consistent training stimulus for reserves in order to maintain the physiological adaptations required to elicit improvements in performance throughout the season. 

It has been previously demonstrated that the accumulative stress of an entire season in NCAA Division I men’s [15] and women’s [16] soccer leads to significant hormonal perturbations, hematological changes and decrements in aerobic fitness and power, despite lower training loads as the season progressed, in relation to the higher workloads during the pre-season period. Further, post-season declines in soccer-specific fitness parameters appear to be related to the amount of match playing time completed by each player throughout the season [17]. When grouped by starter status, starters appear to experience greater reductions in strength, speed and power compared to reserves following a season in men’s collegiate soccer [15]. These trends indicate that players completing higher workloads, which is likely the case in starters, may be at a greater risk for declines in fitness entering the post-season period; however, this has yet to be examined in women’s collegiate soccer. 

Athlete monitoring is particularly useful when comparing one season to another and examining seasonal outcomes in team success, performance levels, fatigue, and injury rates. By monitoring accumulative workloads throughout a season, practitioners can help guide coaching decisions regarding workload management in starters and reserves. Furthermore, this monitoring could better direct conditioning activities during training sessions for reserves if needed. Appropriate workload management may optimize playing performance leading in to post-season play and reduce risks of injuries throughout the season. Therefore, the purpose of this study was to examine differences in accumulated workloads throughout the season between starters and reserves in collegiate women’s soccer.

## 2. Methods

### 2.1. Subjects

Twenty-two NCAA Division III collegiate women soccer athletes (height: 1.67 ± 0.05 m; body mass: 65.42 ± 6.33 kg; fat-free mass: 48.99 ± 3.81 kg; body fat %: 25.22 ± 4.78%) participated in this observational study. Athletes who were medically cleared to participate in practice and match play were eligible to participate in this study. Athletes who were not an active member of the women’s soccer team, or were currently injured at the start of pre-season, were excluded from participation. For the purposes of player status determination, a threshold of >50% of total match duration for the season was used to designate players as starters (*n* = 8) or reserves (*n* = 14) based on previously used methods [13]. Goalkeepers were excluded from this study. All participants provided written consent in accordance to the Institutional Review Board of the University of Wisconsin—La Crosse and Human Subjects Guidelines for Research. 

### 2.2. Study Design

Players were initially invited to an informational meeting prior to the start of the 2019 season during which time details of their participation were explained to them. Demographic information was collected at this time and used to create personalized player profiles in the monitoring system’s software platform. Players were equipped with wearable global positioning systems with on-board inertial sensors which assessed heart rate and movement kinematics when worn. Starting during the pre-season training period, all players wore the GPS-based monitoring units throughout the duration of each practice. Players were instructed to position the monitors in place once they were on the field and about to start the warm-up. The monitors were removed at the end of active practice for a total of 47 practices. Players followed the same protocol during all matches at the start of the competitive season for a total of 22 matches. 977 practice files (367 and 610 from starters and reserves, respectively) and 467 match files (172 and 295 from starters and reserves, respectively) were included in the analysis for a total of 1444 unique player sessions. Workload values were then summed at the end of the season for each of the internal and external load variables recorded by the system.

### 2.3. Athlete Monitoring System

All players were equipped with a GPS-based monitoring system with built-in heart rate monitoring capabilities (Polar TeamPro^TM^ Polar Electro, Oy, Finland). Player demographic information, including age, height and weight, was entered into the proprietary software program associated with the monitoring system which was used to predict aerobic capacity and max heart rate based on age and manufacturer algorithms. The max heart rates (HR) were continually adjusted throughout the pre-season to provide the most accurate and up to date measure of maximal HR. Heart rate zones were used to quantify intensity and defined as: zone 1 = 50–60%, zone 2 = 60–70%, zone 3 = 70–80%, zone 4 = 80–90%, and zone 5 = 90–100%. The software provided a proprietary metric referred to as Training Load which was calculated from heart rate intensity and duration of activity, presented in arbitrary units. At the end of each training session or match, each sensor was removed from the players, loaded to a docking station, and synced to a cloud-based software program operated by the manufacturer. Data were then exported from this program and later used for analysis. 

### 2.4. Movement Kinematics

The monitoring system provided a count of the frequency of accelerations and decelerations using the following thresholds for categorization: low = ±0.5–1.99 m∙s^−2^, moderate = ±2.00–2.99 m∙s^−2^, and high = ±3.00–50.0 m∙s^−2^ based upon previously used methods [13]. The following thresholds were used for determination of speed walk/stand ≤6.99 km·h^−1^, jog = 7.0–14.99 km·h^−1^, run = 15.0–18.99 km·h^−1^, and sprint ≥19.00 km·h^−1^. High speed distance (HSD) was a combination of run and sprint speed zones. Sprints were also counted in an accumulating fashion and were defined as any movement resulting in an acceleration >2.8 m∙s^−2^. For reference, a detailed summary of the match demands by position has been previously published [13].

### 2.5. Statistical Analyses

Differences in accumulated training load and movement characteristics between starters and reserves were examined using a repeated measures analysis of variance. When significant main effects or interactions were identified, Bonferroni post hoc analysis were calculated to determine where differences existed. Normal distribution was confirmed via visual inspection of normal Q-Q plots and via assessment of skewness/kurtosis values. Alpha was set at *p* < 0.05 for determination of statistical significance. Pairwise differences were used to calculate Cohen’s d (d) effect sizes along with 95% confidence intervals (LB, UB) to determine the magnitude of differences in accumulated workload values. The effect sizes were interpreted using the following criteria: 0.2 = trivial, 0.2–0.6 = small, 0.7–1.2 = moderate, 1.3–2.0 = large, and >2.0 = very large [18]. All data were analyzed using IBM SPSS Statistic for Windows (Version 25.0; IBM Corp., Armonk, NY, USA). 

## 3. Results

Accumulated total distance was significantly greater for starters compared to reserves for the total season (starters: 401.7 ± 31.9 vs. reserves: 272.9 ± 51.4 km; *p* < 0.001; d = 2.83 [1.62, 4.03]) and matches (starters: 222.0 ± 21.3 vs. reserves: 100.9 ± 38.6 km; *p* < 0.001; d = 3.61 [2.23, 4.98]) as presented in Figure 1A. Accumulated HSD was significantly greater for starters compared to reserves for all sessions (starters: 38.5 ± 11.9 vs. reserves: 24.8 ± 8.8 km; *p* = 0.006; d = 1.37 [0.41, 2.33]) and matches (starters: 24.3 ± 8.0 vs. reserves: 10.2 ± 6.5 km; *p* < 0.001; d = 2.00 [0.95, 3.05]), as presented in Figure 1B. Accumulated training load was significantly higher in starters compared to reserves for the total season (starters: 9431 ± 1471 vs. reserves: 6310 ± 2263; *p* = 0.002; d = 1.54 [0.56, 2.53]) and matches (starters: 5515 ± 753 vs. reserves: 2392 ± 1217; *p* < 0.001; d = 2.90 [1.68, 4.12]), respectively as presented in Figure 2A. Accumulated number of sprints during matches (starters: 364.3 ± 102.8 vs. reserves: 180.9 ± 78.5; *p* < 0.001; d = 2.09 [1.02, 3.15]) and the season (starters: 700.6 ± 186.5 vs. reserves: 484.9 ± 169.4; *p* = 0.012; d = 1.23 [0.29, 2.17]) was significantly higher for starters compared to reserves, respectively, as presented in Figure 2B. Total distance (starters: 179.7 ± 12.4 vs. reserves: 172.0 ± 25.5; *p* = 0.438; d = 0.35 [−0.52, 1.23]), HSD (starters: 14.2 ± 4.2 vs. reserves: 14.6 ± 3.8; *p* = 0.820; d = 0.10 [−0.77, 0.97]), sprints (starters: 336.4 ± 90.2 vs. reserves: 304.0 ± 103.8; *p* = 0.470; d = 0.33 [−0.55, 1.20]), and training load (starters: 3916 ± 885 vs. reserves: 3918 ± 1358; *p* = 0.998; d = 0.00 [−0.87, 0.87]) did not differ between starters and reserves during practice sessions throughout the season. Significant and meaningful differences in accumulated time spent in the different heart rate zones and acceleration totals for matches throughout the entire season were observed as presented in Table 1, Table 2 and Table 3.

## 4. Discussion

The aim of the current study was to examine differences in accumulated workloads between starters and reserves throughout a collegiate soccer season. Several practically meaningful differences in accumulated measures of external and internal workloads were observed between starters and reserves. Throughout the season, starters covered a greater total distance (401.7 km) and greater distances covered at high speeds (38.5 km) compared to reserves (272.9 and 24.8 km, respectively). Differences in accumulated workload appear to be primarily driven by differences in match playing time, as accumulated match workloads followed a similar pattern as the season totals, whereas few differences in accumulated loads were present when training sessions only were assessed (see Figure 1 and Figure 2). Throughout the season, starters covered a greater distance across all velocity zones, with the exception of sprinting, which was defined as >19.00 km·h^−^^1^ (Table 1). Similarly, starters recorded a greater number of accelerations in all three acceleration categories (Table 1). These differences in external workload elicited similar discrepancies in internal workloads as starters also spent more time in HR zones 3–5 throughout the season. 

When comparing differences in the heart rate-based metric of internal workload (training load), starters again accumulated greater training loads throughout the season compared to reserves. Similar to external workloads, these discrepancies in accumulated measures of internal workload appear to be driven by the differences in match playing time, in which reserves, by definition, did not play as much during match play and therefore slowly accumulated less loads throughout the season. The results of the current study are the first to profile accumulated workloads throughout a season in collegiate women’s soccer. Previously Curtis et al. [14] examined accumulated workloads throughout a season in NCAA Division I men’s collegiate soccer and reported total distances of ~400 km for starters and ~325 km for reserves. Interestingly, the accumulated total distance for the starters is almost identical to that from the current study indicating the men’s and women’s collegiate soccer seasons result in comparable total distances covered, despite differences in the level of play (DI vs. DIII). Additionally, and in alignment with findings from the current study, in Division I men, starters also accumulated a higher number of accelerations, covered greater distances in all velocity zones, spent more time at a heart rate intensity of 70–90% of max heart rate and display greater TRIMP values (measure of internal workload) throughout the season compared to reserves [14]. 

These findings are not surprising considering previous work has indicated comparable match demands between collegiate men and women, regarding distances covered and heart rate response [12,13]. However, the reserves in the study by Curtis et al. [14] accumulated substantially more total distance and TRIMP values throughout the seasons (5 teams monitored over 2 seasons) during training sessions, relative to the starters within the same study. Additionally, the reserves in the current study also did not cover as much distance as those from the study by Curtis et al. [14]. Together, these data suggest that players at the Division I level, may train with greater workloads during practices, or complete a greater number of practices throughout a season compared to Division III. Based on these observations, it appears as though a collegiate men’s and women’s soccer season may otherwise lead to similar total external workloads throughout a season for starters. One notable difference for internal workload, is that women may spend a greater amount of time at >90% HR max (890 min; HR zone 5) compared to men (586 min), which was actually classified as HR zone 6 in the study by Curtis et al. [14]. 

It is difficult to discern whether the observed discrepancies in accumulated workloads throughout the season in the current study are an indication that starters are exposed to excessive workloads or reserves are potentially undertraining throughout the season. A limitation of the current study is that neither indices of performance nor biological parameters of training stress or readiness were assessed throughout the season. Therefore, it is difficult to examine any specific relationships between the workloads in the current study and how they may have impacted player performance, fitness levels, physiological adaptations, or hematological changes throughout the season. However, previous research has indicated that greater accumulated workloads throughout a soccer season lead to greater disturbances in hematological, hormonal and autonomic function, and larger reductions in measures of performance [15,19,20,21]. For example, Walker et al. [21] examined changes in biomarkers throughout a season in collegiate women’s soccer and found that athletes with higher training loads and exercise energy expenditure experienced significant physiological changes throughout the course of the season. Mainly, free and total cortisol, prolactin, triiodothyronine (T3), Interleukin-6 (IL-6), creatine kinase, and total iron-binding capacity levels significantly increased throughout the season. Whereas a significant decrease in Omega-3, iron, hematocrit, ferritin and percent transferrin saturation also occurred. Another important finding of this study is that the magnitude of the physiological changes coincided with higher training loads and exercise energy expenditure during the preseason period, which were further intensified by the cumulative effects of the workloads throughout the rest of the season [21]. In a similar study, Huggins et al. [20] also observed clinically meaningful changes in several hematological, hormonal, and inflammatory biomarkers throughout a collegiate soccer season in men. However, to what extent these changes were influenced by individual player loads were not examined, therefore it is difficult to identify how the magnitude of accumulated workload may have exerted any causal influence on such markers. Saidi et al. [22] analyzed 16 elite Tunisian soccer players were over a 12 week period, assessing them at 3 different times, each following various accumulative workloads. The greatest reductions in sprint performance occurred following the most intensive workload period, which also happened to occur at the end of the season, when the accumulative effects of the season may have been a contributing factor. Such physiological changes should be taken into consideration for starters as they are exposed to greater workloads throughout the season and could potentially experience more perturbations in these biomarkers. 

There is also the concern that a higher accumulation of workloads throughout a season, may predispose starters to a greater risk of injury. While not assessed in the current study, previous work as suggested that excessive workloads may be associated with greater risks for injuries. The challenge lies within the identification of a threshold by which additional workloads may be deleterious to recovery and performance; and how to identify such a threshold before the line is crossed. Following a 2 year monitoring period in elite youth soccer, authors noted greater relative risks of overall and non-contact injuries following periods of higher accumulative workloads, particularly over 3 and 4 week periods of higher total distances and higher number of accelerations [23]. Similarly, in a study examining weekly training and match loads of 46 elite Australian footballers, the authors reported that as the workload increased, so did the risk of injury. The authors, therefore, recommended that the practice of monitoring and adjusting workloads weekly, may help reduce the risk of injury [24]. These findings are supported by an extensive review conducted by Eckard et al. [25], in which the authors concluded that based on the current evidence there appears to be a strong relationship between training load and injury. Because starters are faced with greater exposures and playing times, they are inherently at a great risk for injury. This elevated risk may be greater at the collegiate level as there is a higher frequency of competition compared to the professional level; however, more evidence is needed to support this hypothesis. 

On the contrary, the reduced accumulative workload experienced by the reserves, may also be an area of concern as it may lead to detraining effects; however, previous work in this area has yielded mixed results. At the collegiate [26] and professional level [27], it has been demonstrated that starters were able to maintain and even increase various measures of performance throughout a basketball season, with performance increases tending to coincide with increased playing time. Thereby indicating that regular and sufficient stimulus throughout the season may help elicit performance adaptations throughout the season. Contradictory findings were noted by McLean et al. [28], who reported significant decline in maximal power output during inertial load cycling in starters throughout the season in soccer players, but not in non-starters. In, an earlier study by Kraemer et al. [15], the authors noted significant reductions in vertical jump and sprint performance at the end of a soccer season in starters, which were not observed in non-starters. The authors [15] concluded that because of higher accumulative workloads throughout the season, the starters may have been overtrained, which may have subsequently contributed to the observed reductions in performance. Therefore, it may be a mixture of the two in that the higher accumulative workloads experienced by starters may provide a continued stimulus for further improvements throughout the season but excessive amounts may result in an overtraining effect, in which, athletes may experience decrements in performance and other disturbances in biomarkers. Whereas non-starters may benefit from additional training throughout the season, in order to continue providing regular stimuli likely required to elicit an adaptive response to compensate for the lack of match play. The lack of in-season and post-season performance assessments is a limitation of the current study design. Without these performance assessments, it is difficult to determine if the starters were exposed to excessively high workloads or conversely, if the reserves were not exposed to sufficient workloads required to elicit physiological adaptations or maintain fitness status throughout the season. Future research should employ regular in-season performance testing to evaluate fitness levels, neuromuscular function and overall player readiness to help guide decision making on load management strategies and player readiness.

## 5. Conclusions

The total accumulative distance covered by starters was almost double that covered by reserves throughout the season. Further, starters accumulated greater HSD, higher training loads, and a greater number of sprints throughout the season for all sessions compared to reserves. Significant discrepancies were also observed for the amount of time starters and reserves spent in different HR zones. Specifically during match play, starters spent most of the match in HR zone 4, while reserve players were found to spend most of their time in HR zone 1, which also contributed to greater accumulative internal workloads (training load) throughout the season for starters. Higher workloads have been linked to more accentuated perturbations of several biomarkers when compared to lower workloads, which may also carry a higher risk of injury. Discrepancies in accumulated workloads throughout the season between starters and reserves have been previously established in men’s, and now women’s collegiate soccer. However, there have been mixed findings regarding how such workload discrepancies influence the ability to maintain fitness levels throughout a season. Future studies should seek to examine which end of the spectrum is more problematic: the higher workload experienced by starters or the lower workload experienced by reserves. Both extremes may be detrimental to players’ health and performance. On one hand, the higher load experienced by starters, may lead to overtraining, accumulative fatigue, inflammation, higher risk of injury and psychological disturbances. On the other hand, the lower load experienced by reserve players may not be sufficient to maintain fitness status throughout a season and would therefore would warrant additional training for reserves throughout the season.

## Figures and Tables

**Figure 1 jfmk-07-00011-f001:**
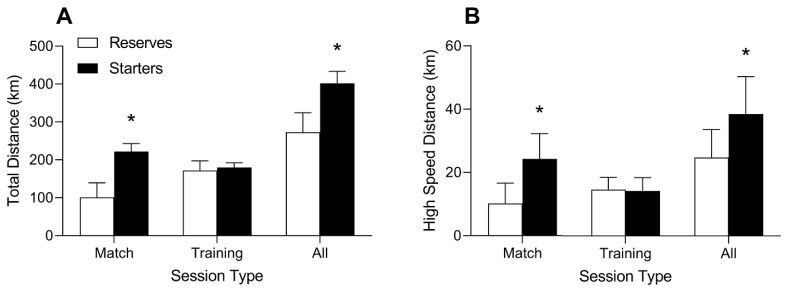
Differences in total distance covered (**A**) and high speed distance (**B**) between starters and reserves in match settings, practice settings, and the cumulative total season. * *p* < 0.001.

**Figure 2 jfmk-07-00011-f002:**
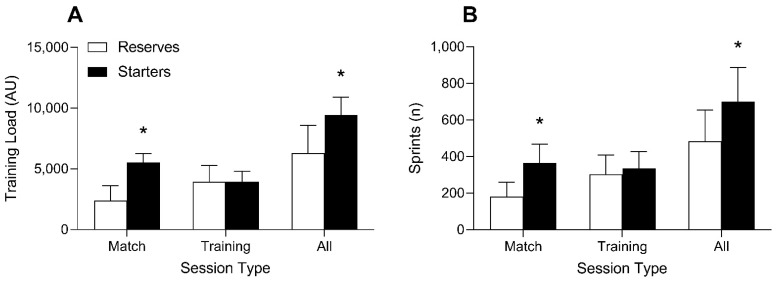
Differences in accumulated training load (**A**) and number of sprints (**B**) between starters and reserves in match settings, practice settings, and the cumulative total season. * *p* < 0.001.

**Table 1 jfmk-07-00011-t001:** Total accumulated workloads for all sessions in NCAA Division III women’s soccer by starting status (mean ± SD).

Variable	Starters	Reserves	*p*	Cohen’s d (LB, UB)
VZ1 (km)	188.8 ± 16.4	139.5 ± 23.7	<0.001	2.30 (1.20, 3.40)
VZ2 (km)	174.4 ± 29.6	108.6 ± 24.0	<0.001	2.52 (1.38, 3.67)
VZ3 (km)	27.4 ± 7.3	17.4 ± 5.8	0.002	1.58 (0.59, 2.56)
VZ4 (km)	11.0 ± 5.4	7.4 ± 3.2	0.059	0.89 (−0.02, 1.80)
HRZ1 (min)	1460 ± 278	1473 ± 217	0.906	0.05 (−0.82, 0.92)
HRZ2 (min)	1362 ± 253	1258 ± 386	0.508	0.30 (−0.57, 1.17)
HRZ3 (min)	1161 ± 263	920 ± 255	0.048	0.93 (0.02, 1.85)
HRZ4 (min)	1346 ± 369	870 ± 275	0.003	1.53 (0.55, 2.51)
HRZ5 (min)	923 ± 576	511 ± 251	0.029	1.04 (0.12, 1.96)
AZ1 (n)	88,234 ± 8807	65,991 ± 10,932	<0.001	2.17 (1.09, 3.25)
AZ2 (n)	5674 ± 858	3743 ± 985	<0.001	2.05 (0.99, 3.11)
AZ3 (n)	1011 ± 295	728 ± 241	0.023	1.09 (0.16, 2.01)

Velocity Zone Abbreviations: VZ1 (Walk/Stand) ≤6.99 km·h^−1^; VZ2 (Jog) = 7.0–14.99 km·h^−1^; VZ3 (Run) = 15.0–18.99 km·h^−1^; VZ4 (Sprint) ≥19.00 km·h^−1^. Heart rate zone abbreviations: HRZ1 = 50–60%; HRZ2 = 60–70%; HRZ3 = 70–80%; HRZ4 = 80–90%; HRZ5 = 90–100%. Acceleration Zone Abbreviations: AZ1 = ±0.5–1.99 m·s^−2^; AZ2 = ±2.00–2.99 m·s^−2^; AZ3 = ±3.00–50.0 m·s^−2^.

**Table 2 jfmk-07-00011-t002:** Total accumulated match-day workloads in NCAA Division III women’s soccer by starting status (mean ± SD).

Variable	Starters	Reserves	*p*	Cohen’s d (LB, UB)
VZ1 (km)	92.9 ± 12.3	48.8 ± 13.7	<0.001	3.34 (2.02, 4.65)
VZ2 (km)	104.8 ± 19.3	41.9 ± 19.5	<0.001	3.24 (1.95, 4.53)
VZ3 (km)	17.7 ± 5.2	7.4 ± 4.5	<0.001	2.14 (1.06, 3.21)
VZ4 (km)	6.6 ± 3.5	2.7 ± 2.0	0.004	1.46 (0.49, 2.43)
HRZ1 (min)	505 ± 178	677 ± 178	0.042	0.96 (0.05, 1.88)
HRZ2 (min)	506 ± 91	418 ± 183	0.221	0.56 (−0.32, 1.44)
HRZ3 (min)	490 ± 146	281 ± 96	0.001	1.81 (0.79, 2.83)
HRZ4 (min)	842 ± 368	286 ± 161	<0.001	2.19 (1.11, 3.28)
HRZ5 (min)	700 ± 403	269 ± 157	0.002	1.60 (0.61, 2.58)
AZ1 (n)	42,659 ± 5305	23,771 ± 6356	<0.001	3.14 (1.87, 4.41)
AZ2 (n)	3069 ± 545	1358 ± 660	<0.001	2.75 (1.56, 3.94)
AZ3 (n)	529 ± 162	268 ± 125	<0.001	1.87 (0.84, 2.90)

Velocity Zone Abbreviations: VZ1 (Walk/Stand) ≤6.99 km·h^−1^; VZ2 (Jog) = 7.0–14.99 km·h^−1^; VZ3 (Run) = 15.0–18.99 km·h^−1^; VZ4 (Sprint) ≥19.00 km·h^−1^. Heart rate zone abbreviations: HRZ1 = 50–60%; HRZ2 = 60–70%; HRZ3 = 70–80%; HRZ4 = 80–90%; HRZ5 = 90–100%. Acceleration Zone Abbreviations: AZ1 = ±0.5–1.99 m·s^−2^; AZ2 = ±2.00–2.99 m·s^−2^; AZ3 = ±3.00–50.0 m·s^−2^.

**Table 3 jfmk-07-00011-t003:** Total accumulated practice session workloads in NCAA Division III women’s soccer by starting status (mean ± SD).

Variable	Starters	Reserves	*p*	Cohen’s d (LB, UB)
VZ1 (km)	95.9 ± 4.9	90.8 ± 14.8	0.352	0.42 (−0.46, 1.30)
VZ2 (km)	69.5 ± 11.6	66.6 ± 11.0	0.567	0.26 (−0.61, 1.13)
VZ3 (km)	9.7 ± 2.4	10.0 ± 2.5	0.850	0.08 (−0.78, 0.95)
VZ4 (km)	4.5 ± 2.0	4.7 ± 1.6	0.797	0.12 (−0.75, 0.98)
HRZ1 (min)	955 ± 151	796 ± 134	0.019	1.13 (0.20, 2.06)
HRZ2 (min)	856 ± 177	841 ± 226	0.872	0.07 (−0.80, 0.94)
HRZ3 (min)	671 ± 154	639 ± 182	0.686	0.18 (−0.69, 1.05)
HRZ4 (min)	504 ± 134	583 ± 187	0.306	0.47 (−0.41, 1.35)
HRZ5 (min)	223 ± 201	241 ± 154	0.810	0.11 (−0.76, 0.98)
AZ1 (n)	45,576 ± 4104	42,220 ± 6427	0.201	0.59 (−0.30, 1.47)
AZ2 (n)	2606 ± 414	2385 ± 524	0.320	0.45 (−0.43, 1.33)
AZ3 (n)	482 ± 148	459 ± 142	0.723	0.16 (−0.71, 1.03)

Velocity Zone Abbreviations: VZ1 (Walk/Stand) ≤6.99 km·h^−1^; VZ2 (Jog) = 7.0–14.99 km·h^−1^; VZ3 (Run) = 15.0–18.99 km·h^−1^; VZ4 (Sprint) ≥19.00 km·h^−1^. Heart rate zone abbreviations: HRZ1 = 50–60%; HRZ2 = 60–70%; HRZ3 = 70–80%; HRZ4 = 80–90%; HRZ5 = 90–100%. Acceleration Zone Abbreviations: AZ1 = ±0.5–1.99 m·s^−2^; AZ2 = ±2.00–2.99 m·s^−2^; AZ3 = ±3.00–50.0 m·s^−2^.

## Data Availability

The data presented in this study are available on request from the corresponding author.

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
