# Peer review of "Seasonal Accumulated Workloads in Collegiate Women’s Soccer: A Comparison of Starters and Reserves"

_jfmk, 2022, doi:10.3390/jfmk7010011_

Round 1

Reviewer 1 Report

Abstract

  • Clear and informative.

Introduction

  • Line 64, perhaps break into a new paragraph.
  • Line 75, perhaps break into a new paragraph.

Methods

  • I have a Team Pro system. For training background, which one did you click? It seems, given the differences between starters and non, they actually might not have the same training background.
  • Any of the other possible values entered apart from what you wrote?
  • Line 149 – why not repeated measures MANOVA?
  • Line 157 – any need to reference Cohen's d interpretations? Or do you feel these are common enough and basically facts?

Results

  • Outstanding tables and figures. Thank you.

Discussion

  • Lines 209-253, perhaps a few paragraphs rather than one long one, will help the readers.
  • Line 324 – You make the point. Perhaps you can make it stronger or not. Or perhaps the point is teams should do sub-maximal or some sort of fitness tests monthly to help understand what is going on with such large training differences.

Reviewer 2 Report

The aim of the study was to examine differences in accumulated workloads throughout the season between starters and reserves in collegiate women’s soccer. 

The research is innovative and it was carried out methodologically correctly.

The work is written following the steps of the scientific method.

In the introduction, I propose to write a bit more about training loads, how to measure loads. It is worth paying more attention to reactions of the body to these loads.

Add inclusion and exclusion issues to your study.

Possibly add limitation for the performance of the study. 

The conclusions from the study may be more interesting. They should be written again, giving specific results of own research, which clearly result from the tables.

The article is generally valuable and correctly written, please treat the above comments only as suggestions. 
